# Cholera past and future in Nigeria: Are the Global Task Force on Cholera Control's 2030 targets achievable?

**Gina E. C. Charnley** [1,2]*, **Sebastian Yennan**[3], **Chinwe Ochu**[3], **Ilan Kelman**[4,5,6], **Katy A. M. Gaythorpe**[1,2], **Kris A. Murray**[1,2,7]

**1** Department of Infectious Disease Epidemiology, School of Public Health, Imperial College London, London, United Kingdom, **2** MRC Centre for Global Infectious Disease Analysis, School of Public Health, Imperial College London, London, United Kingdom, **3** Surveillance and Epidemiology Department/IM Cholera, Nigeria Centre for Disease Control, Abuja, Nigeria, **4** Institute for Risk and Disaster Reduction, University College London, London, United Kingdom, **5** Institute for Global Health, University College London, London, United Kingdom, **6** University of Agder, Kristiansand, Norway, **7** MRC Unit The Gambia at London School of Hygiene and Tropical Medicine, Fajara, The Gambia

* g.charnley19@imperial.ac.uk

**Data Availability Statement:** All data used here are taken from public sources and referenced throughout, except the cholera data which was provided by the NCDC and contains data potentially

## Abstract

### Background

Understanding and continually assessing the achievability of global health targets is key to reducing disease burden and mortality. The Global Task Force on Cholera Control (GTFCC) Roadmap aims to reduce cholera deaths by 90% and eliminate the disease in twenty countries by 2030. The Roadmap has three axes focusing on reporting, response and coordination. Here, we assess the achievability of the GTFCC targets in Nigeria and identify where the three axes could be strengthened to reach and exceed these goals.

### Methodology/Principal findings

Using cholera surveillance data from Nigeria, cholera incidence was calculated and used to model time-varying reproduction number (R). A best fit random forest model was identified using R as the outcome variable and several environmental and social covariates were considered in the model, using random forest variable importance and correlation clustering. Future scenarios were created (based on varying degrees of socioeconomic development and emissions reductions) and used to project future cholera transmission, nationally and sub-nationally to 2070. The projections suggest that significant reductions in cholera cases could be achieved by 2030, particularly in the more developed southern states, but increases in cases remain a possibility. Meeting the 2030 target, nationally, currently looks unlikely and we propose a new 2050 target focusing on reducing regional inequities, while still advocating for cholera elimination being achieved as soon as possible.

### Conclusion/Significance

The 2030 targets could potentially be reached by 2030 in some parts of Nigeria, but more effort is needed to reach these targets at a national level, particularly through access and

identifiable and sensitive patient information and is therefore available on request after signing a data sharing agreement with NCDC. The data can be requested via the Nigeria Centre for Disease Control and the data here were obtained from the Surveillance and Epidemiology Department/IM Cholera. The email for NCDC is: info@ncdc.gov.ng and the institutional website is: https://ncdc.gov.ng.

**Funding:** This work was supported by the Natural Environmental Research Council [NE/S007415/1] (GECC), as part of the Grantham Institute for Climate Change and the Environment's (Imperial College London) Science and Solutions for a Changing Planet Doctoral Training Partnership. We also acknowledge joint Centre funding from the UK Medical Research Council and Department for International Development [MR/R0156600/1]. The funders had no role in study design, data collection and analysis, decision to publish, or preparation of the manuscript and no authors received a salary from any of your funders.

**Competing interests:** The authors declare no competing interests.

incentives to cholera testing, sanitation expansion, poverty alleviation and urban planning. The results highlight the importance of and how modelling studies can be used to inform cholera policy and the potential for this to be applied in other contexts.

## Author summary

Using a random forest model and future scenarios to project cholera risk, we assessed the achievability of the Global Task Force for Cholera Control 2030 Roadmap in Nigeria, based on sustainable development and environmental protection. The results highlighted how regional inequities within Nigeria will likely prevent the 2030 targets being reached. The northern states in Nigeria are more rural, less developed and have greater levels of insecurity and conflict. Reaching the levels of peace and development achieved in the south at a national level, along with improving urban planning and access and incentives to cholera testing, will be fundamental in reaching the Roadmap targets in Nigeria. Cholera predictions and projections are understudied and here we present the most in-depth projections currently produced for Nigeria. Projections provide a snapshot of potential future conditions and show what is needed going forward to meet and exceed health targets and reduce disease burden. We highlight how quantitative research can be used to inform policy and the relevance of doing so. Quantitative research should fundamentally aim to improve global health and by presenting our work with a strong policy focus, we highlight the relevance and importance of doing so.

## Introduction

Global health and development targets are a set of goals which can be used to identify best strategies to address challenges and provide a common target or path for governments and organisations to work towards. In 1992, the Global Task Force on Cholera Control (GTFCC) was established as a global partnership of more than 50 institutions (including governments, academic institutions, non-governmental organisation and UN agencies) and in 2017, the GTFCC launched "Ending Cholera: A Global Roadmap to 2030". The Roadmap encouraged partner organisations to sign the Declaration to End Cholera, which focused on three axes: 1. early detection and response, 2. interventions in cholera hotspots and 3. effective coordination at all levels. The GTFCC aims to stop country-wide uncontrolled cholera outbreaks by 2030 and eliminate the disease from twenty countries, resulting in a 90% reduction in cholera deaths [1].

Despite the commitment of governments and organisations to large-scale strategies, many argue that these goals and targets fail to produce significant improvements [2,3]. There have been gains in cholera control at a local level in recent decades, through widespread implementation of cholera prevention and education by sub-national health ministries, although this has been minimal at the global level [4,5]. As the 2030 goal approaches, whether these targets can be achieved given the current pace of development and progress is highly uncertain [6]. Understanding the achievability of these targets will help countries and global partnerships to plan for 2030 and beyond, continuing to make improvements in cholera prevention and control. Goals should be ambitious and encourage partners to strive for the best outcome possible, but they also need to be clear and have significant commitment and motivation from governments, non-governmental organisations and the population.

There is an estimated 1.3 billion people at risk of cholera globally and approximately 2.86 million annual cases (1.3–4.0 million) [7,8], the majority of which are in sub-Saharan Africa and the Indian Subcontinent. Development and health are strongly connected, and several countries and regions with the lowest levels of development, in terms of poverty, education and health, also overlap with areas of high cholera burden [9]. Nigeria currently has one of the largest cholera burdens globally and a significant number of people living in poverty, making it a critical area of study. Nigeria is also one of the largest African economies, has the largest populations, and has made several gains in socioeconomic development in recent decades [10].

Here, we aim to understand if the GTFCC 2030 targets are achievable in Nigeria, based on the current Roadmap and progress. To understand if Nigeria is on track to meet these 2030 targets, we project cholera transmission (using cholera time-varying reproduction number (R)) to 2070 both nationally and sub-nationally (administrative level 1). The projections are based on scenarios of varying degrees of global change using a random forest model, proved robust at predicting R in previous studies [11]. Relatively few studies have projected cholera burden, most of which have taken a climate change focus [12–14] and even fewer studies have evaluated the likelihood of meeting the 2030 GTFCC targets using model projections [15,16]. The projection results will then be used to inform the three axes outlined in the Roadmap, highlighting both successes and areas for improvement. Finally, we make suggestions of how to reach the 2030 targets in Nigeria, suggesting specific thresholds, triggers and areas for prioritisation.

## Methods

### Ethics statement

The datasets and methods used here were approved by Imperial College Research Governance and Integrity Committee (ref: 22IC7711) and a data sharing agreement between the Nigeria Centre for Disease Control (NCDC) and the authors. Formal consent was not obtained for individuals in the data used here, as the data were anonymized before publication.

### Datasets

Cholera data were obtained from NCDC and contained surveillance data for 2018 and 2019. Each datapoint represented one case and included information on the date of symptom onset (DD-MM-YY), location (administrative level 4, village/settlement), if the case was confirmed, age, sex and outcome of infection (hospitalisation and death/recovered). The data were subset to only include cases that were confirmed either by rapid diagnostic tests or by laboratory culture and only these confirmed cases were used (which is standard when modelling R [17]). A sensitivity analysis, aimed at understanding if removing suspected cases bias the results, is presented in S1 Information. Additionally, NCDC provided oral cholera vaccination (OCV) data. The data were represented by the campaign start and end date, the location (administrative level 1) and the coverage. OCV was transformed to an annual binary outcome variable (0–1) for each state (e.g., if coverage was 100% in a specific year and state, the data point was assigned 1).

A range of covariates were considered in the best fit model based on previously understood cholera risk factors and are present in Table 1. Multiple drought metrics were used, measured across different time windows, this was used to investigate both relative dryness/wetness, not long-term drought changes, due to the short timescale of the cholera surveillance dataset. For model fitting, administrative level one (state) was set as the spatial granularity (data on a finer spatial scale were attributed to administrative level 1) and the finest temporal scale possible for

**Table 1. The environmental and social data considered in the best fit model.**

| Covariate | Source | Temporal Scale | Spatial Scale |
|---|---|---|---|
| Palmers Drought Severity Index (PDSI) | National Center for Atmospheric Research [18] | Monthly | Administrative level 1 |
| Standardised Precipitation Index | Centre for Environmental Data Analysis [19] | Monthly | Administrative level 1 |
| WASH (improved drinking water, piped water, improved sanitation, open defecation, basic hygiene, annual) | Joint Monitoring Programme [5] | Annual | Geopolitical zone |
| Internally Displaced Persons (households and individuals) | International Organization for Migration [20] | Annual | National |
| Healthcare (total facilities, facilitates per 100,000 population) | Humanitarian Data Exchange [21] | Annual | Administrative level 1 |
| Poverty (Multi-Dimensional Poverty Index (MPI), headcount ratio in poverty, intensity of deprivation among the poor, severe poverty and population vulnerable to poverty) | Humanitarian Data Exchange [21] | Annual | Administrative level 1 |
| Conflict events and fatalities | Armed Conflict Location & Event Data Project (ACLED) [22] | Daily, Monthly | Administrative level 3 |
| Population | United Nations Department for Economic and Social Affairs [23] | Annual | Administrative level 1 |

covariate selection. The granularity chosen best captured the range in the available data and values were repeated where needed for data on a finer temporal scale.

WorldClim [24] was used for the projected temperature and precipitation data, at administrative level 1 and a monthly temporal granularity. WorldClim data is from Coupled Model Intercomparison Project 6 (CMIP6) downscaled future gridded temperature and precipitation projections (aggregated to administrative level 1), processed for nine global climate models. The data included minimum temperature and maximum temperature measured in degrees Celsius (°C) and precipitation (in mm). Projections were single values for 2050 and 2070 using three different Representative Concentration Pathways (RCPs) (RCP4.5, 6.0 and 8.5).

## Cholera incidence and reproduction number

The confirmed cholera surveillance data for 2018 and 2019 were used to calculate daily incidence by taking the sum of the cases reported by state and date of onset of symptoms. R was calculated from incidence using a serial interval (SI) of 5 days, with a standard deviation of 8 days [25–29]. R is a measure of disease transmission and provides information on outbreak evolution e.g., if R is greater than 1, cases are likely to increase.

Estimating R when incidence is low over the time window, increases error. Coefficient of variation (CV), at a threshold of 0.3 (or less) as standard, based on previous work [26], was used to reduce uncertainty in periods of low incidence. To reach the CV threshold, R values were calculated over monthly sliding windows, calculation start date for each state was altered where necessary and states with <40 cases were removed (as states with fewer cases did not have high enough incidence across the time window to reach the CV threshold) (packages "incidence" [25] & "EpiEstim" [26]).

## Covariate selection and model fitting

The 22 covariates in Table 1 were first clustered to prevent multicollinearity in the final model. The clustering was based on corrections between the covariates meeting an absolute pairwise correlation of above 0.75. Random forest variable importance was used to rank the clustered covariates. Variable importance is a measure of the cumulative decreasing mean standard error each time a variable is used as a node split in a tree.

Training (70% of data) and testing (30%) datasets were created to train the regression random forest model and test the model's predictive performance. The parameters for training were set to repeated cross-validation for the resampling method, with ten resampling interactions and five complete sets of folds to complete. The model was tuned and estimated an optimal number of predictors at each split of 2, based on the lowest out-of-bag error rate with Root-Mean-Square Error (RMSE) used as the evaluation metric (package "caret" [30]).

A stepwise analysis was used to fit the models to cholera R, taking into consideration the covariate clustering and variable importance. One covariate was selected from each cluster, and all combinations of covariates were tested until the best-fit model was found. Models were assessed based on predictive accuracy using correlations between testing and training, coefficient of determination ($R^2$) and RMSE. To test for over-fitting (fitting to the testing dataset too closely or exactly), mean absolute error (MAE) was calculated on the training R values against the predicted testing R values, little to no error in the predictions are an indication of over-fitting. All analysis was completed in R version 4.1.0, and additional information on the model, including sensitivity analyses are available in a complementary paper [11].

## Scenarios and projection

Five projection scenarios were created here to project cholera to 2070 using the best fit model. Despite 2030 being the target year for the Roadmap, cholera projections were made to the furthest point the data allowed (2070), at decadal time steps. The scenarios include several degrees of global change and are based upon the RCP scenarios and attainment of the Sustainable Development Goals (SDGs) [31]. By using the RCPs and SDGs, this covered a wide range of environmental and social scenarios, accounting for varying degrees of emissions reductions and socioeconomic development. Both the RCPs and SDGs are widely accepted methods and targets for environmental and socio-economic development in research and policy. The scenarios are briefly defined as follows, and more details are given below:

1. Scenario 1 (S1)–"Best-case" scenario meeting RCP4.5 and the SDGs

2. Scenario 2 (S2)—Intermediate progress scenario between S1 and S3

3. Scenario 3 (S3)—Minimal development and emissions reductions but progress is still made towards to SDGs and the Intergovernmental Panel on Climate Change targets (RCPs)

4. Scenario 4 (S4)—Some regression from the current levels of sustainable development and increased emissions

5. Scenario 5 (S5)–"Worst-case" scenario with significant regression in development and emissions increases

Sub-national monthly PDSI projections were calculated using the projected 2050 and 2070 environmental data. First, potential evapotranspiration (PET) (mm/day) was calculated with the temperature data and latitude using the Hargreaves method [32], where $R_a$ is the mean extra-terrestrial radiation in mm/day, which is a function of latitude, and T represents daily air temperature in ˚C (package "SPEI" [33]). PDSI is the output of a supply-and-demand model of soil moisture, which can be calculated using PET and precipitation. The calculations provided PDSI values for 2050 and 2070 for the three RCP scenarios to administrative level 1 (package "scPDSI" [34]).

$$PET_{hargreaves} = 0.0023*R_a*(T_{max} - T_{min})^{0.5}*(T_{mean} + 17.8).$$

Poverty scenarios were set at varying degrees of achieving SDG1.1 and 1.2 in Scenarios 1–3,

**Table 2. Cholera projection scenarios for 2030–2070 at decadal intervals.**

| Scenario | Year | PDSI | MPI | Sanitation | Conflict |
|---|---|---|---|---|---|
| Scenario 1 | 2030 | 2020 value | 50% decrease | 50% increase | 50% decrease |
| | 2040 | | | | |
| | 2050 | RCP4.5 2050 | Median 2040–2070 | Median 2040–2070 | Median 2040–2070 |
| | 2060 | Median 2050–2070 | | | |
| | 2070 | RCP4.5 2070 | Elimination | 100% access | Elimination |
| Scenario 2 | 2030 | 2020 value | 2020 value | Median 2020–2050 | 2020 value |
| | 2040 | | | | |
| | 2050 | RCP6.0 2050 | 50% decrease | 50% increase | 50% decrease |
| | 2060 | Median 2050–2070 | Median 2050–2070 | | Median 2050–2070 |
| | 2070 | RCP6.0 2070 | Elimination | | Elimination |
| Scenario 3 | 2030 | 2020 value | 2020 value | 2020 value | 2020 value |
| | 2040 | | | | |
| | 2050 | RCP8.5 2050 | | Median 2040–2070 | |
| | 2060 | Median 2050–2070 | Median 2050–2070 | | Median 2050–2070 |
| | 2070 | RCP8.5 2070 | 50% decrease | 30% increase | 50% decrease |
| Scenario 4 | 2030 | RCP6.0 2050 | 2020 value | 2020 value | 2020 value |
| | 2040 | | Median 2030–2070 | | Median 2030–2070 |
| | 2050 | RCP6.0 2070 | | Median 2040–2070 | |
| | 2060 | | | | |
| | 2070 | | 30% increase | 30% decrease | 30% increase |
| Scenario 5 | 2030 | RCP8.5 2050 | 2020 value | 2020 value | 2020 value |
| | 2040 | | Median 2030–2070 | | Median 2030–2070 |
| | 2050 | RCP8.5 2070 | | Median 2040–2070 | |
| | 2060 | | | | |
| | 2070 | | 50% increase | 50% decrease | 50% increase |

the goal aims for a 50% reduction in extreme (<1.25/day) poverty by 2030 and poverty eliminated by 2070. Whereas in S4 and S5, there was a 30% and 50% increase in poverty, respectively. The sanitation scenarios were based on SDG6.2 (achieve access to adequate and equitable sanitation and hygiene for all and end open defecation by 2030) and conflict scenarios were guided by SDG16.1 (significantly reduce all forms of violence and related death rates everywhere). The SDGs for both sanitation and conflict are particularly ambiguous, regardless of this difficulty, the sanitation and conflict targets were based on a similar pattern to the poverty scenarios (MPI), achieving universal access to sanitation and conflict elimination by 2070 in S1 and a 50% decrease in sanitation access and 50% increase in conflict events by 2070 in S5. A summary of the scenarios are shown in Table 2.

Both national and subnational projections were calculated, taking an average of the subnational values for each scenario and year for the national projections (or taking the national value, where reported). Bootstrap resampling (10,000 samples) was used to obtain 95% confidence intervals for all projections. The projections were applied to the 2030 Roadmap targets and a new 2050 target delineated. Twenty-fifty was chosen as the new proposed target year, due to it being halfway from the current target (2030) to the end of the projection period (2070) and the results suggesting that this may be achievable. Additionally, the two global initiatives the scenarios were based upon, the RCPs and SDGs, have new mid-century targets [35,36].

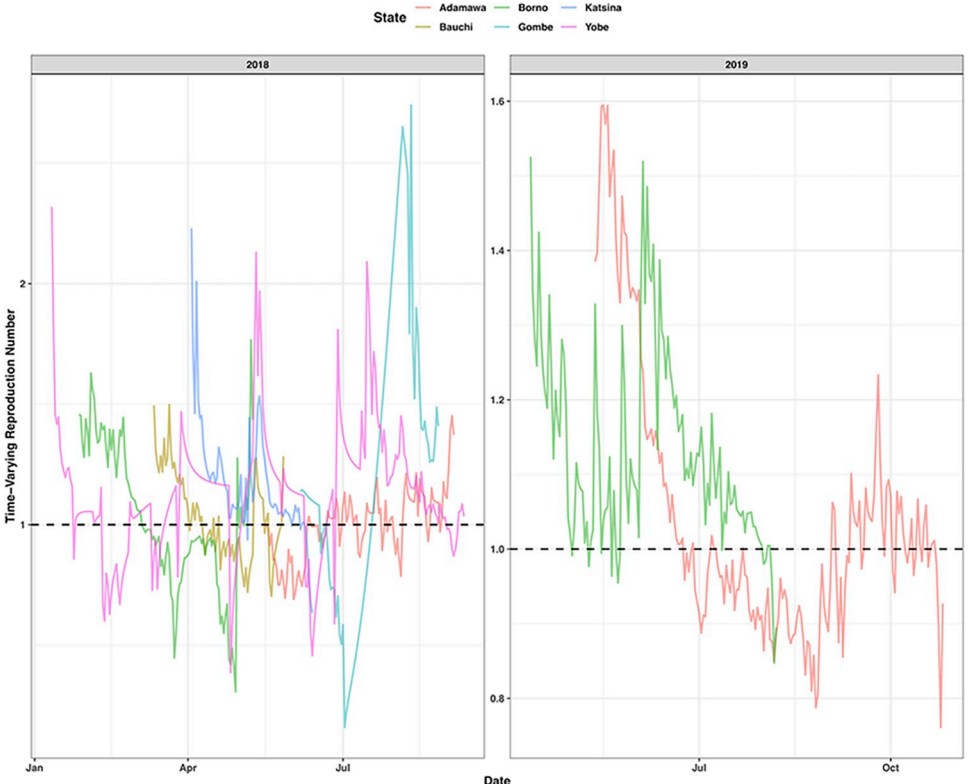

**Fig 1. Cholera time-varying reproduction number (R) for the six states in Nigeria (Adamawa, Borno, Katsina, Bauchi, Gombe and Yobe) which met the criteria for inclusion in the R calculations for 2018 and 2019.** The black dashed line shows the R threshold for cases increasing (R >= 1) or cases decreasing (R < 1).

## Results

### Covariate selection and best fit model

In Nigeria, 1,401 cholera cases were confirmed in 2018 and 2019 (out of 46,694 suspected and confirmed). The results from the sensitivity analysis including confirmed and suspected cases, proved model robustness and that the smaller dataset was not biasing the results (S1 Information). The geographic distribution of confirmed cases was concentrated in the northeast of the country, with Adamawa, Borno, Katsina and Yobe having the highest burden. Six states for 2018 and two states for 2019 had sufficient cases to be included for R calculations (Fig 1), including Adamawa (2018 & 2019), Bauchi (2018), Borno (2018 & 2019), Gombe (2018), Katsina (2018) and Yobe (2018).

The twenty-two covariates were grouped according to correlation into nine clusters. The clusters and variable importance (based on reducing node impurity) of each covariate are shown in Fig 2. Stepping through model possibilities of the covariates, the best fit model included number of monthly conflict events, Multidimensional Poverty Index (MPI) (annual), Palmers Drought Severity Index (PDSI) (monthly) and improved access to sanitation (annual), fitted to R values with a correlation of testing predictions to training values of 0.87, RMSE at 0.17 and $R^2$ of 0.51.

### Scenario projections

The national scenario projections saw only marginal changes in R over the projection period (2030–2070) (Fig 3), potentially due to the national averaged values reducing uncertainty and

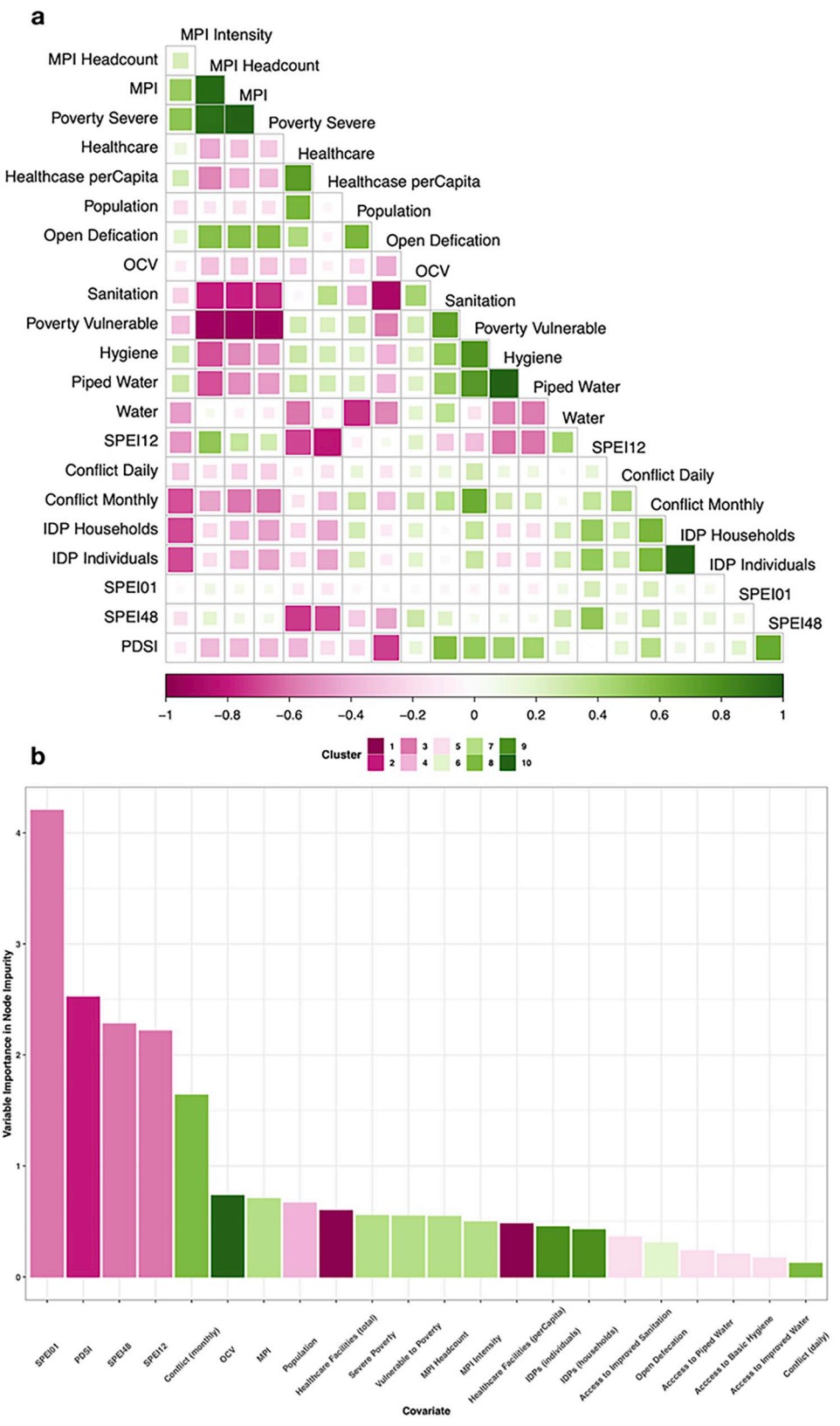

**Fig 2. Environmental and social covariates considered in the best fit model. a**, a correlation matrix of the absolute pairwise correlations for the covariates, -1 indicate a negative correlation and +1 a positive correlation. **b**, random forest variable importance based on node impurity of the covariates and their clusters based on the correlations from **a**.

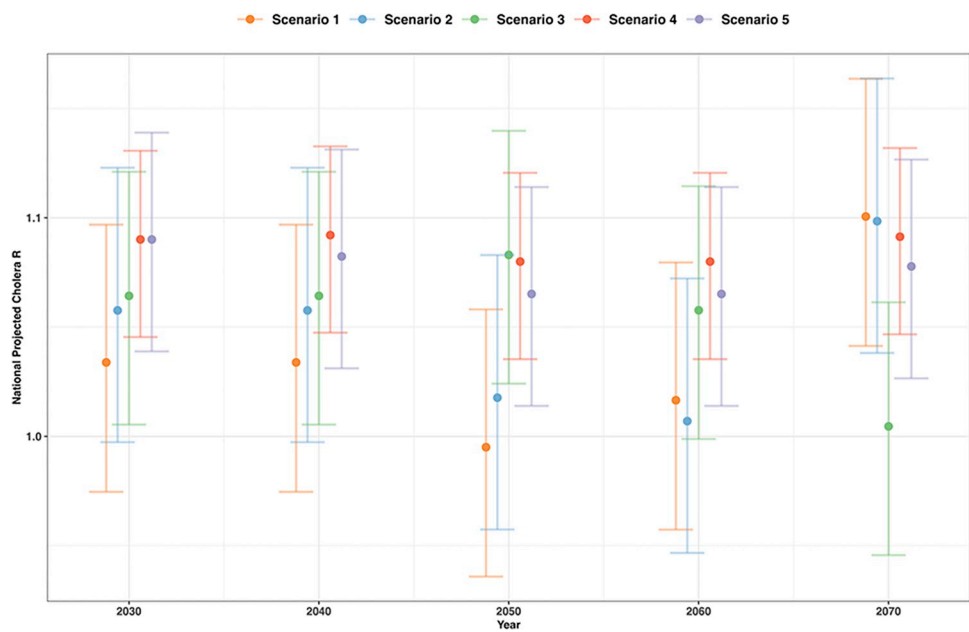

**Fig 3. National cholera projections for Nigeria, in cholera reproduction number (R), for the five scenarios (S1—orange, S2—blue, S3—green, S4—red and S5—purple) at 3 of the decadal time points (2030, 2050 & 2070) with 95% confidence intervals.**

therefore smoothing out any heterogeneity. The relatively wide uncertainty overlaps for all scanrios and only Scenario 1 sees R values below 1. The greatest decrease in R was projected in S1, followed by S2, however values then increased for 2060 and 2070. For S3, there was minimal change with the values fluctuating around the 2030 value. S4 and S5 were similar to S3 with only minimal changes, but there was a slight increase in cholera R with worsening environmental and social conditions.

For the sub-national projections, measured in terms of cholera R, there were several spatial heterogeneities, which may help explain some of the uncertainty seen in the national projections. Generally, R values decreased through the three time points shown (Fig 4) for S1, S2 and S3, with the number of states with R values over 1 decreasing, particularly for S1 to S3. However, like the national analysis, some states increased from 2050 to 2070, this is likely due to changing environmental conditions, which had very high variable importance here (see Fig 3) and therefore only small differences in PDSI resulted in projection changes. For example, the increase in some states in S3, from 2050 to 2070, is due to PDSI projections being closer to 0 in 2050 and then decreasing in 2070, with both extreme wet and dry being important for cholera transmission in the model [11]. For S4 and S5, some states appeared to fare better than others when faced with worsening social and environmental conditions. The south of the country had particularly high R values in these less optimistic scenarios, whereas the north saw little change and, in some cases, a slight improvement. Despite the heterogeneity, by 2050 average projected R values for most regions (based on Nigeria's six geopolitical zones) in Nigeria had R value projections of less than 1 in Scenario 1, lending to the new proposed 2050 target (S1 Fig).

## Discussion

The aim of the research presented here was to use a random forest model to make cholera projections to 2070 and use these projections to understand if and when Nigeria could meet the GTFCC 2030 targets. Each country committed to the GTFCC Roadmap has their own unique

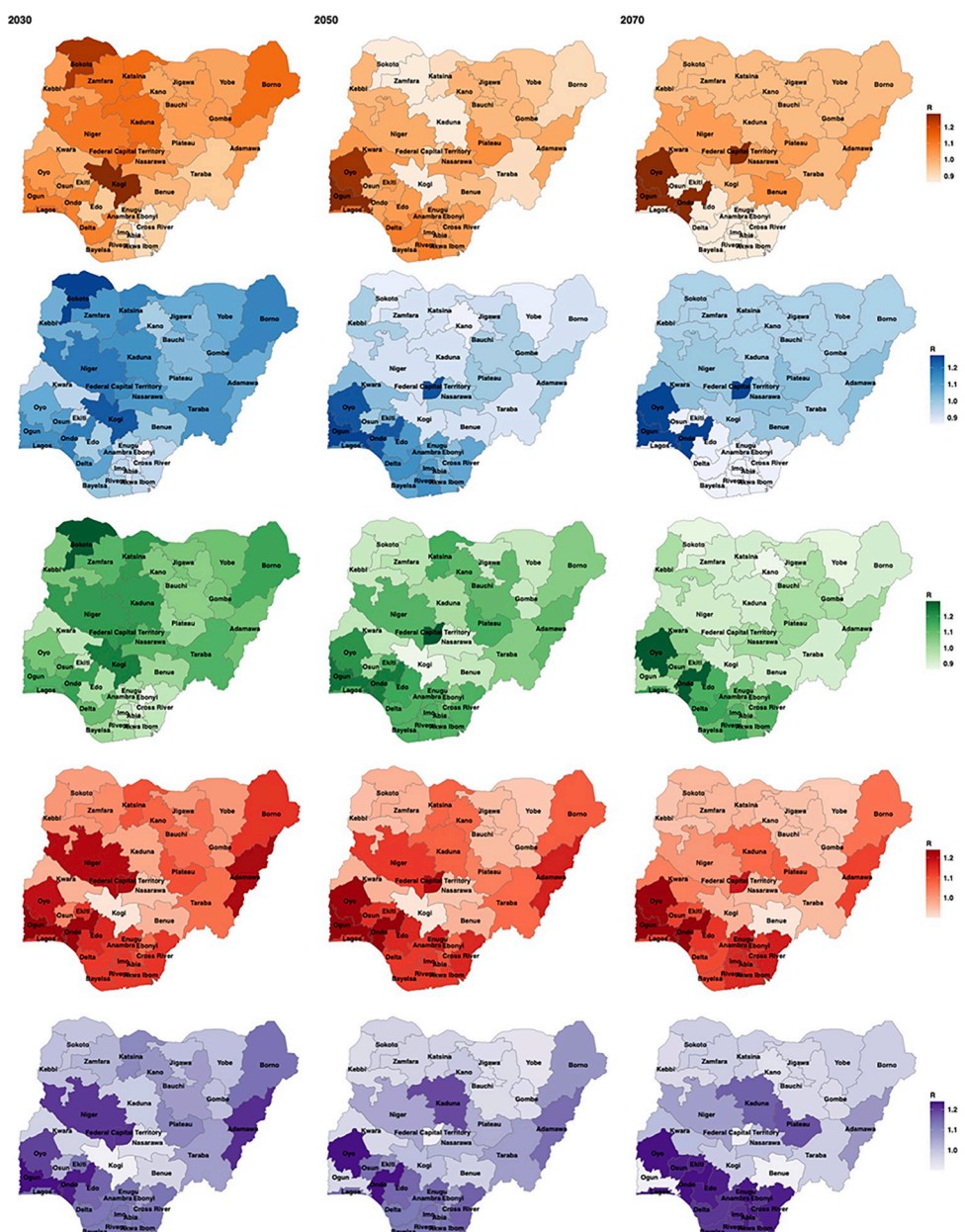

**Fig 4. Sub-national cholera projections for Nigeria, in cholera reproduction number (R), for the five scenarios (S1 —orange, S2—blue, S3—green, S4—red and S5—purple) at 3 of the decadal time points (2030, 2050 & 2070).** License: CC-BY [37].

set of challenges and priorities and here we aimed to use quantitative research to help tailor the GTFCC Roadmap to Nigeria. The best fit model included conflict event frequency, PDSI, MPI and percentage access to improved sanitation. The model was then used to make cholera projections and provided a more detailed understanding of future trends and if socio-economic development and climate change mitigation could reduce cholera in Nigeria. Both the national and sub-national projections showed decreases in cholera transmission with the more optimistic scenarios (S1-S3), however at a national level, these declines in transmission where only marginal and unlikely to result in the decreased mortality needed to meet the 2030 targets.

Under S4 and S5, cholera transmission worsened, despite the already high levels (R > = 1) in Nigeria, showing the need for continued development.

Several poverty and WASH covariates were clustered together during covariate selection and included in the best fit model. Cholera is considered a disease of inequity and global maps of cholera burden often track with areas of high poverty. Cholera is a water-borne disease, and access to sanitation and clean water, which are often lacking in areas of high poverty, is fundamental in cholera transmission [38]. Progress has been made in the last 20 years (2000–2020) in terms of expanding access to WASH services in Nigeria, with the percentage of access to improved sanitation increasing from 52% to 62% and an expansion in access to improved drinking water by 31% [5]. However, if access to sanitation continues to increase by 10% every 20 years, this would result in over 87 million Nigerians without access by 2030 and therefore at a high risk of cholera (based on a 5% increase in access and a population projection of 264 million by 2030 [23]).

In Nigeria, there is a divide in terms of WASH and development between the northern and southern regions of the country (S2 Fig) [39]. Northern states are generally more rural, which has potentially resulted in less development. For example, there is a 40% deficit in access to sanitation in the rural compared to the urban population. Nigeria's rural population comprises 47% (99,895,289, 2021) of the total population, putting millions of people at risk of cholera in these potentially less developed areas [40]. In the last sixty years the rural population has decreased by 38%, if this trend continues, effective urban planning is essential to offset cholera risk, as densely populated urban areas, with poor climate resiliency and provision of services e.g., sanitation, healthcare, infrastructure, have proved a risk for cholera outbreaks [41].

## Evidence from the national projections

The national scenarios showed clear trends in terms of both improvements from S1 (lowest cholera transmission) to S3 and regression in S4 and S5. However, there were some increases in cholera R after 2060, with wide ranging uncertainty throughout, potentially due to residual confounding. The projections showed that continued progress towards and beyond the SDGs (in particular SDG1, 6 and 16) and emissions reductions (contributing to extreme PDSI and temperature) would help to improve global health and particularly cholera. However, by 2030, in all the national projection, cholera was not close to the 2030 targets (90% reduction). By the end of the projection period (2070), with significant improvements in development and environmental protection, cholera transmission was still relatively high and far from eliminated. Cholera eradication will likely take time, due to the pathogen circulating in the population and environmental reservoirs and via introductions of new cases/strains due to travel [42,43] and continued global commitment is needed.

## Evidence from the sub-national projections

The sub-national projections were more optimistic regarding the achievability of the GTFCC targets, due to the heterogeneity across the country. Overall, for the S1 sub-national projections there was a decrease in R values to less than 1 by 2050 for nearly all regions and for the southern states all S1 cholera projections were less than 1 by 2030 (compared to the northern states which were all above 1). An explanation for this is that more time will be needed in the northern states to reach the required development for a significant decline in transmission. Additionally, in the northern states, development was already low and conflict already high, therefore worsening these conditions did not significantly impact cholera.

The projections suggest that southern Nigeria could potentially reach the 2030 targets and eradication may be possible in the future. The north of the country must be an area of

prioritisation in terms of development, cholera response and conflict resolution. The sub-national projections additionally highlighted how vital it will be for the development and peace achieved in the south to continue or at a minimum, remain the same. As previously stated, the worsened conditions of S4 and S5 had a large impact on the southern states in terms of increasing cholera transmission. Decreasing levels of peace and development would potentially be catastrophic in the southern states and overall, to the cholera burden in the country.

## An update on the 2030 targets and roadmap

Based on the work presented here, the achievability of the GTFCC targets at the current pace of cholera control and development appears unlikely to be met by 2030 in Nigeria. Despite this, there has been significant progress in terms of health and development in Nigeria and the southern states appear far more likely to meet these goals. A new proposed 2050 target, building on the GTFCC Roadmap and expanding on the three axes will now be delineated, while still advocating that the targets are reached as soon as possible. The new targets will aim at bringing the northern states to the same levels of development and peace achieved in the south.

## Axis 1

Axis 1 largely focuses on surveillance and data, which are vital in target setting, allocation of resources and response. Improved surveillance may also help to reduce the lack of cholera data used here (only two years) and additionally testing would help to improve the validity of the data available (only 1,401 cholera cases were confirmed out of 46,694 suspected). The axis focuses on early detection, but with cholera this is difficult, due to a large number of mild infections and several other diarrhoeal pathogens potentially causing disease (e.g., shigella, typhoid, dysentery), meaning a positive test does not always mean that cholera is the causative agent of the clinical signs.

A method to help improve reporting in Nigeria, would be to offer incentives to test and report. Financial incentives have proved effective at improving health outcomes in Nigeria and are often cost-effective in the long-term as they prevent serious disease and morbidity [44,45]. To reduce nosocomial transmission and prevent testing hesitancy due to stigmatisation and restrictions [16], modification and improvements to the highly effective cholera rapid diagnostic test [46], allowing them to be used at home, may be helpful. The tests would need to be easy to use and report, inexpensive and widely available.

Emphasis is needed both at a government and academic level on improving data quantity and quality. Understanding reporting effort and the accuracy and precision of data are key areas of future research to fully understand how well the current data are representing cholera burden. At a global level, a metric of reporting effort would help when comparing disease data that has been collected across multiple countries and therefore with different methods and uncertainty and similar metrics have already been developed [47].

Furthermore, risk factor data are needed to fully understand disease dynamics and plan for effective response and interventions. The covariates selected in the best fit model here suggest that improvements in tracking poverty and sanitation are good areas of prioritisation. In Nigeria, benefit could also be gained from testing environmental reservoirs, such as major lakes and rivers, which are known to be used for washing and drinking and can be fundamental in cholera transmission [48,49]. Better understanding on the environmental burden would be useful for both research and to understand local risk factors.

## Axis 2

Axis 2 (cholera interventions) is arguably the most important area for reaching the GTFCC goals and several other health targets. The Roadmap highlights the need for long-term sustainable WASH implementation and strengthening of healthcare systems to anticipate cholera outbreaks (e.g., capacity building of staff, resources, diagnostics, education and societal engagement and emergency WASH intervention). However, the GTFCC Roadmap and previous research on cholera interventions heavily focuses on outbreak response [15], rather than development.

The Roadmap suggests that interventions should target states most at risk, with the analysis presented here suggesting northern states as a priority. Additionally, healthcare in Nigeria should be strengthened more generally, with greater resources and service availability, making healthcare an attractive career option to ensure sufficient human resources [50,51]. Development planning and targets must also consider that global crises can cause regression of progress, increasing the need to strive beyond health targets. For example, COVID-19 is estimated to have erased four years of progress against poverty and caused disaster-related deaths to rise sixfold [31].

Designating significant financial resources on outbreak response, is not a cost-effective way of reaching cholera targets, although fundamental to reducing mortality in outbreaks. More emphasis needs to be placed on improving peoples' quality of life, lifting them out of poverty, providing them with basic services and empowering them to improve their own health through resources and education. For example, healthcare spending in Nigeria is currently at a level not seen since 2002, at 3.03% of Gross Domestic Product. Only five countries in Africa spend less on healthcare than Nigeria and health needs to be a greater priority in terms of policy and government spending [52]. In the absence of this development, outbreaks will continue to occur, spending financial resources in a reactionary way.

## Axis 3

Axis 3 of the Roadmap involves commitment and coordination on a global level, across several sectors. NCDC currently works across multiple levels of the national systems and has a detailed response plan for diarrhoeal (including cholera) outbreaks titled, "Preparedness and Response to Acute Watery Diarrhoea Outbreaks" [53]. NCDC have a designated team working on cholera elimination as a priority within the country. Continued and increased funding to NCDC will be vital for them to continue their work toward cholera control. At a global level, a "One World—One Health" approach is needed to prevent pandemics and achieve the GTFCC targets at this level. Recent pandemics and global outbreaks (e.g., COVID-19 and mpox) have shown the catastrophic results of countries not working together in a joint effort to control disease [54].

Nigeria has made several gains in weakening the Boko Haram stronghold in the northeastern states. However, the conflict continues to threaten Nigeria's security and several previous studies have suggested the negative impacts of conflict on health [55–57]. Bottom-up stabilisation efforts are working to address local level drivers of insecurity, including strengthening local conflict prevention, restoring governance and services and fostering social cohesion. At a regional level, the Lake Chad Basin Commission and African Union Commission have highlighted short-, medium- and long-term stabilisation, resilience and recovery needs [58–60]. Fig 5 illustrates the current Roadmap and summarises the suggestion made here to improve cholera control beyond 2030 and achieve the GTFCC targets in Nigeria by 2050.

| Axis 1<br>Early detection and response | Axis 2<br>Multisectoral approach | Axis 3<br>Mechanism of coordination |
|---|---|---|
| • Early warning surveillance systems<br>• Pre-positioning stocks<br>• Preparedness of WASH systems<br>• Preparedness of the healthcare system<br>• Community engagement | • Identify hotspots and priority areas<br>• Control measures (surveillance, WASH, health care systems, OCV, community engagement and collaboration) | • Nationally-led cross-sectoral programs<br>• GTFCC as a strong coordination platform |
| • Incentives to report<br>• Rapid at-home tests<br>• Monitoring of the environment and known risk factors | • Shift from outbreak response to long-term sustainable development<br>• Empower local populations to care for their health through poverty alleviation, services and education<br>• Effective urban planning | • Global commitment and coordination for a "One World - One Health" approach to pandemics<br>• Conflict resolution<br>• Increase health expenditure |
| 2030 | 90% reduction in deaths and no large-scale outbreaks | |
| 2050 | Targeting the northern states and continued development in the south | |

**Fig 5. The 2030 GTFCC Roadmap for cholera elimination (black) with additional suggestions and areas of prioritisation (blue) for 2050 in Nigeria.**

## Limitations

Data incompleteness and inconsistencies were issues when finding data sources and fitting the models here. Incomplete data resulted in either the analysis not being completed (e.g., data fit to models before 2018) or removing and averaging the data (both used in the data fitted to the model). As stated above, improving surveillance and a greater effort to collect data on cholera risk factors will be very important for target setting and resource allocation and prevent duplication of services [61], along with aiding and improving scientific research into cholera dynamics. In future cholera research, using sensitivity analysis and testing cholera assumptions across multiple data sources is one method to understand these differences.

All scenario projections have limitations, due to the uncertainty in trying to predict future conditions, along with the limitations of models. The wide range of future scenarios helps to account for some of this uncertainty but will still not be sufficient in capturing all potential future environments. For example, the scenarios here are uni-directional, either getting better or worse from current conditions. All social and environmental drivers either getting better or worse is unlikely, with some metrics improving and some worsening. To add further complexity, these changes could be spatially heterogeneous. Regardless of these limitations, this should not discourage scenario projection analysis, as it is still useful and valid in terms of understanding future changes from previous patterns and relationships and help to inform cholera prevention and policy.

A further limitation is the outcome variable of cholera transmission, whereas the GTFCC targets largely focus on reducing cholera deaths. Although the discrepancies create difficulties in comparing the projection results to the GTFCC targets, to reach the 2030 targets and subsequently reduce deaths by 90%, global cholera transmission will have to substantially decrease, regardless of the metric used. Therefore, the projections are still useful in presenting the required decrease in cholera needed.

## Conclusions

In Nigeria, the GTFCC targets look difficult to achieve by 2030 at a national level, based on these results. However, the more urban and developed southern states may reach and go

beyond these targets. There is a vital need for continued investment in long-term development, especially to reduce regional inequity in northern Nigeria. Despite the financial capital needed to improve healthcare, WASH and education, these interventions are cost-effective due to their wide-reaching impacts.

The results presented here highlight the importance of and how modelling studies can be used to inform cholera policy and the potential for this to be applied to other diseases and countries. The aim of health research should be to improve global health and quality of life and applying quantitative research to policy can help achieve this, while increasing the relevance of the research. Using quantitative research to identify specific thresholds and areas of prioritisation can help to focus disease control efforts and prevent action fatigue.

If the GTFCC targets are met in Nigeria, this will reduce the risk of cholera for hundreds of millions of people and greatly reduce the global burden of diarrhoeal disease and mortality, particularly in the most vulnerable. Progress to go beyond the goals and targets set would be highly beneficial in combating the impact of global shocks and crisis. Continued progress in development (especially at an accelerated pace) and sustainable urban planning, could greatly improve Nigeria's cholera burden and health status in the coming decades.

## Supporting information

**S1 Information. Sensitivity analysis using confirmed and suspected cholera cases.** The analysis includes R calculations, variable importance and model fitting for the full dataset. (DOCX)

**S1 Fig. Sub-national projected changes in cholera transmission (R) for Nigeria.** Average regional R value for each scenario at 2050. The regions are based on the six Nigerian geopolitical zones. North Central: Benue, Kogi, Kwara, Nasarawa, Niger, Plateau, Federal Capital Territory. North East: Adamawa, Bauchi, Borno, Gombe, Taraba, Yobe. North West: Jigawa, Kaduna, Kano, Katsina, Kebbi, Sokoto, Zamfara. South East: Abia, Anambra, Ebonyi, Enugu, Imo. South Central: Akwa Ibom, Bayelsa, Cross River, Delta, Edo, Rivers. South West: Ekiti, Lagos, Ogun, Ondo, Osun, Oyo. (TIFF)

**S2 Fig. Development indicators in Nigeria.** Average values for the full dataset by state, for **A**, percentage access to sanitation and **B**, Multidimensional Poverty Index (MPI). The sources and timescales of the data are shown in Table 1. License: CC-BY, available from: https://data.humdata.org/dataset/cod-ab-nga. (TIFF)

## Acknowledgments

We would like to thank and acknowledge all those who collected and curated the datasets used here, including the Nigeria Centre for Disease Control for their direct involvement.

## Author Contributions

**Conceptualization:** Gina E. C. Charnley, Ilan Kelman, Katy A. M. Gaythorpe, Kris A. Murray.

**Data curation:** Sebastian Yennan, Chinwe Ochu.

**Formal analysis:** Gina E. C. Charnley.

**Funding acquisition:** Gina E. C. Charnley.

**Investigation:** Gina E. C. Charnley, Sebastian Yennan, Chinwe Ochu, Ilan Kelman, Katy A. M. Gaythorpe, Kris A. Murray.

**Methodology:** Gina E. C. Charnley, Ilan Kelman, Katy A. M. Gaythorpe, Kris A. Murray.

**Supervision:** Ilan Kelman, Katy A. M. Gaythorpe, Kris A. Murray.

**Validation:** Sebastian Yennan, Chinwe Ochu, Katy A. M. Gaythorpe, Kris A. Murray.

**Visualization:** Gina E. C. Charnley.

**Writing – original draft:** Gina E. C. Charnley, Ilan Kelman, Katy A. M. Gaythorpe, Kris A. Murray.

**Writing – review & editing:** Gina E. C. Charnley, Ilan Kelman, Katy A. M. Gaythorpe, Kris A. Murray.

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
