## [Decision Letter · Decision Letter 0]

16 Jan 2023

Dear Dr Charnley,

Thank you very much for submitting your manuscript "Cholera past and future in Nigeria: are the Global Task Force on Cholera Control’s 2030 targets achievable?" for consideration at PLOS Neglected Tropical Diseases. As with all papers reviewed by the journal, your manuscript was reviewed by members of the editorial board and by several independent reviewers. In light of the reviews (below this email), we would like to invite the resubmission of a significantly-revised version that takes into account the reviewers' comments. 

Each of the three reviewers expressed significant concerns about the public health significance or methodology described for modeling and forecasting. Thus, the manuscript will require considerable revision before it is returned to the reviewers for evaluation. Please read the reviewer comments carefully and if you choose, revise the manuscript to incorporate or rebut their comments.

We cannot make any decision about publication until we have seen the revised manuscript and your response to the reviewers' comments. Your revised manuscript is also likely to be sent to reviewers for further evaluation.

Sincerely,

Richard A. Bowen

Academic Editor

Dileepa Ediriweera

Section Editor

Reviewer's Responses to Questions

**Key Review Criteria Required for Acceptance?**

**Methods**

-Are the objectives of the study clearly articulated with a clear testable hypothesis stated?

-Is the study design appropriate to address the stated objectives?

-Is the population clearly described and appropriate for the hypothesis being tested?

-Is the sample size sufficient to ensure adequate power to address the hypothesis being tested?

-Were correct statistical analysis used to support conclusions?

-Are there concerns about ethical or regulatory requirements being met?

Reviewer #1: This is an exploratory study, and not testing any hypothesis. Although, study design is not a problem, there are several methodological issues as given below:

1. The study used the death rates in the analysis. Death is mostly depended on the health system of the country that includes health infrastructure and communication system. Therefore, it could describe the status of the health system of the place, but I do not think analyzing the deaths could predict cholera situation in a country.

2. Cholera death data were taken from two open access sources. It is not clear how the data from the two sources were reconciled. Also, the study used the data until 2016. Since cholera situation changes over time, not including the most recent data set (2017-2022) may yield incorrect forecasting. 

3. The study used the projected climatic data of 2050 and 2070 from WorldClim. There could be several human interventions in future that could change the climatic condition of an area. 

4. The definition of the cholera outbreak occurrence is not clear. Do they define it by at least one reported case? 

5. There is no clarity in the method section about the unit of analysis and how many units were there to analyze the data. Different environmental and social data were collected at different geographic scales. How the data at different geographic scales were processed for fitting in the model? 

6. As I see the covariates were selected using a bivariate (should NOT call call univariate) regression model where the covariates were found to be significantly associated with the outcome variable at a 10% confidence limit. Next, the authors said, “The best fit model was identified based on Bayesian Information Criterion and area under the receiver operator characteristic curve.” It is not clear how the different models were created from which the best model was derived?

7. I don’t see validation of the forecasting model. Without validating the forecasting model, how one could forecast cholera situation in 50 years from now.

8. In Figure S4, it is said that most of the correlations between the covariate and the WHO cholera deaths are significant, potentially due to a lack of complete data. What is meant by lack of complete data. When the data is incomplete what will be the implication of the findings of this study? Also, how do the authors know that the cholera death data are underreported. I am also wondering why a national average was taken from the available data points and used for all years (and not the average of year before and year after) when they found missing data of a year. The method used for interpolating the data would dilute the year wise variation in the data. 

9. It is said that five projection scenarios were created to project cholera to 2070 using the two models. Which specific model was used for which scenario?

Reviewer #2: The main points of the methods get lost in the detail. There are a lot of different models noted in the methods section, but in the introduction it indicates that only two different models were used. It is unclear what the two different models are. Is it a GLM and random forest models? Or is it two of the same model type with different outcome measures? Is it national and sub-national? What is the purpose of the two different model types in answering the research question? There needs to be a more explicit explanation of what models were used (potentially visualized in a table or figure), the variables included in each model, and how each model is used to answer the research question.

Some other specific recommendations:

94 – More comparable to what? Other countries? To compare death rates across different regions of Nigeria? 

96 – “Cholera death data were taken from two open access sources.” is a full sentence. There should be a period at the end not a comma. 

163-164 – This should be split into two sentences.

182-196 – Which SDG’s is this referring to? All of them? Just the ones related to climate? 

206 – Is there research to support this assumption? This seems like a significant assumption to make. Water withdrawl does not indicate who is using the water (or that water access is increasing for everyone). It may mean more water collected by specific subsets of the population but not for others. Water withdrawl also does not specify what the water is used for. More water may be withdrawn during a drought scenario to support increased need for agriculture, not necessarily consumption.

223-224 – This is not a full sentence. 

258-259 – It would be helpful to highlight the differences in the data collection methods that may explain why the data from these two sources is so different. Were they surveying similar study populations? Where they looking at different indicators of cholera mortality (laboratory confirmed vs. reported)?, etc.

Reviewer #3: See comments below

**Results**

-Does the analysis presented match the analysis plan?

-Are the results clearly and completely presented?

-Are the figures (Tables, Images) of sufficient quality for clarity?

Reviewer #1: The results are presented according to the analysis plan. However, since there are problems in the data, I do not think the findings of the study have any implication in real life.

Reviewer #2: Some minor changes to tables and figures may help with the clarity of the results:

Fig 4 – The unit on the y axis should be explained better. In the methods it indicates that outbreak occurrence of 1 indicates at least one cholera case but the interpretation of values lower than 1 should be explained.

Figure 5 – The color scale on this is confusing. The cut toff point of 1 is important for for the purposes of these results, but it is difficult to tell specific values with a continuous color scale. Would a categorical color scale showing <1,1 (+/-0.1),>1 be possible to more clearly visualize reductions and improvements?

Reviewer #3: Yes

**Conclusions**

-Are the conclusions supported by the data presented?

-Are the limitations of analysis clearly described?

-Do the authors discuss how these data can be helpful to advance our understanding of the topic under study?

-Is public health relevance addressed?

Reviewer #1: The conclusions are supported by the results and limitations of the study are briefly described. However, I am afraid the findings of the study have any public health relevance.

Reviewer #2: Many of the points in the discussion and conclusion are limited by either lack of support by the results presented in this analysis or by previous literature:

346-354 – It needs to be stated more explicitly here why sanitation access is a concern for cholera transmission. Explain how limited sanitation access increases risk of cholera transmission (then later why this justifies improvements in environmental sampling). The correlation analysis alone also cannot prove that limited sanitation is the reason for cholera outbreaks because the analysis does not account for potential confounding by other variables (i.e. poverty). The language here needs to be careful to not draw causal conclusions outside of the scope of the analysis.

361-363 – Is this saying that reducing the size of rural populations would reduce cholera outbreaks. What about the increased risk of cholera transmission in more dense urban settings? What is meant by “effective urban planning”? Is this just in terms of WASH access or is this also referring to increased climate resiliency, etc.?

374 – It’s unclear what “and via introductions” means.

376-382 – The conclusions about sub-national cholera projections are all be based on S1 and are ignoring the other scenarios. The broad conclusion that southern states will do better and northern states will do worse does not hold true for S3-S5. It seems like a big stretch to conclude that the sub-national projections are more optimistic when all scenarios show variation across different regions (with some areas having increases in cholera and some having decreases based on the scenario).

379-381 – This statement doesn’t align with what is shown in Figure 5. Northern states showed an initial reduction in 2050 but then a slight increase in 2070. Based on what is shown in the figure, more time would not lead to a reduction in transmission.

403-422 – It is unclear how the data presented in this analysis supports a need for more testing. While this is an important strategy to pursue to collect better data for the GTFCC roadmap, it is outside of the scope of this analysis. It needs to be more explicitly stated why the results from this analysis lead to this recommendation.

442-444 – This sentence is conflicting with the previous paragraph which suggests that the GTFCC roadmap should shift from outbreak response to development. Strengthening healthcare would potentially improve outbreak response but not necessarily development. Strengthening healthcare is also an overly broad recommendation. What specifically about healthcare needs to be improved? This, again, seems to fall outside the scope of this analysis. This claim needs to be supported by either the data presented in this analysis or by previous literature.

449-454 – All sentences in this paragraph need references. 

463-464 – Reference? What is this approach and how does it relate to this analysis?

In general, it is unclear how the recommendations to the GTFCC axes are specifically expected to achieve the goal if 90% reduction by 2050, and, more importantly, how the evidence from this analysis support the recommendations. The data from this analysis shows that none of the projections would lead to a 90% reduction in Cholera, even by 2070, so how is it expected that these recommendations would lead to improvements by 2050? The methods used in this analysis are not able to draw conclusions about causality (i.e. what specifically is causing the increases or reductions in cholera). It seems to stretch outside of the scope of the analysis to recommend specific changes given the lack of support from this analysis about what specifically is causing increases or reductions in cholera.

Reviewer #3: See below. There are suggested changes that will increase the readability for a broader audience.

**Editorial and Data Presentation Modifications?**

Reviewer #1: (No Response)

Reviewer #2: Other suggestions for the introduction section:

46 – Targets are not a type of strategy for addressing challenges, they are a set of goals that can be used to identify best strategies for addressing challenges. The language here could be more precise.

48 – What types of institutions are included? Are these academic? Governmental? NGO?, etc.

58 – Language here is confusing. Is “they” referring to the goals? Or the organizations?

60 – “Gains in cholera control” is ambiguous. Does this mean the prevalence of cholera has gone down? Or that strategies to reduce cholera have been more widely implemented and/or have been more effective? Does “at the local level” mean in specific regions of Nigeria? Or throughout all of Nigeria? More specificity in what progress has been made would be helpful here.

82 – Would be helpful to briefly explain here what the two models are and how they are used to answer the research question.

Reviewer #3: (No Response)

**Summary and General Comments**

Reviewer #1: This study attempted to forecast cholera situation in Nigeria for almost 50 years from now using historical data of cholera death from two different sources. I find this an ambitious project as this study used the data of the last 40-50 years when I do not think there was systematic disease surveillance for keeping the records of morbidity and mortality in that country. The nature of data could result in incompleteness, inconsistency, and incorporating falls positivity that could mislead forecasting of the cholera situation in the country. Therefore, I do not think the outcomes of this study have any practical implacability.

Reviewer #2: The methods appear to be thorough and the results have potential public health relevance, but there are significant limitations in the writing and interpretation of the results. Several points in the discussion section are not supported by evidence from the analysis or other relevant literature. To be ready for publication, this paper either needs an adjustment to the language and interpretations or additional analyses to support the conclusions about causes of cholera and intervention recommendations.

Reviewer #3: The authors have submitted an interesting manuscript questioning the likelihood of Nigeria reaching its 2030 goals regarding cholera control. They have used the WHO Global Health Observatory and the Global Health Data Exchange is their primary sources for cholera mortality and have used WorldClim for climate information. In addition, subnational data were also used to analyze regional differences within Nigeria. By most possible scenarios, including the most likely, the 2030 goals will not be met.

Here are a few comments and questions:

1. Please give more details regarding the data sources, including their methods and potential pitfalls

2. It was not clear what data have been used to estimate its impact on cholera mortality, including how much of the impact was directly related to available potable water.

3. What are the confounders for the use of data in a linear manner, especially as it relates to accuracy of data acquisition.

4. Also, mortality may change in the setting of the same case load if access to or quality of health care changes.

5. It appears that the GHDx data begin in 1990, right where there is a major peak in the WHO data. First, what is the impact for the trend if the GHDx data begin at the peak, which is the highest rate for any of the WHO data? Second, do you have any way to reconcile the differences in data between the two data sources.

The potential role of immunization should be addressed to a greater extent since there is good evidence from Bangladesh that there is herd protection from immunization that could affect the incidence of cases (see Chowdhury, CMR, 2022).

There are also numerous small grammatical issues that must be corrected. A few are shown below:

Line 242 – Do the authors mean defecation rather than defection?

Lines 261, 385, 473 – These sentences begin with a conditional conjunction (whereas, while) that requires the conditional component. As written, they are incomplete sentences.

PLOS authors have the option to publish the peer review history of their article (what does this mean?). If published, this will include your full peer review and any attached files.

Reviewer #1: Yes: Mohammad Ali

Reviewer #2: No

Reviewer #3: No
---

## [Decision Letter · Decision Letter 1]

15 Apr 2023

Dear Dr Charnley,

We are pleased to inform you that your manuscript 'Cholera past and future in Nigeria: are the Global Task Force on Cholera Control’s 2030 targets achievable?' has been provisionally accepted for publication in PLOS Neglected Tropical Diseases.

Best regards,

Richard A. Bowen

Academic Editor

Dileepa Ediriweera

Section Editor

Reviewer's Responses to Questions

**Key Review Criteria Required for Acceptance?**

**Methods**

-Are the objectives of the study clearly articulated with a clear testable hypothesis stated?

-Is the study design appropriate to address the stated objectives?

-Is the population clearly described and appropriate for the hypothesis being tested?

-Is the sample size sufficient to ensure adequate power to address the hypothesis being tested?

-Were correct statistical analysis used to support conclusions?

-Are there concerns about ethical or regulatory requirements being met?

Reviewer #1: (No Response)

Reviewer #2: Objectives were clear and simplified methods, compared to the original submission, were much more straightforward and approachable. Some concerns related to data availability (i.e. line 142 removing states with <40 cases), but these limitations are noted in the discussion section.

Reviewer #3: (No Response)

**Results**

-Does the analysis presented match the analysis plan?

-Are the results clearly and completely presented?

-Are the figures (Tables, Images) of sufficient quality for clarity?

Reviewer #1: (No Response)

Reviewer #2: Results are documented well and figures are used well to explain the findings the analysis.

Reviewer #3: (No Response)

**Conclusions**

-Are the conclusions supported by the data presented?

-Are the limitations of analysis clearly described?

-Do the authors discuss how these data can be helpful to advance our understanding of the topic under study?

-Is public health relevance addressed?

Reviewer #1: (No Response)

Reviewer #2: The conclusions are well explained and are justified based on the analysis presented. Limitations of the analysis are noted while also acknowledging hte public health relevance of the data presented.

Reviewer #3: (No Response)

**Editorial and Data Presentation Modifications?**

Reviewer #1: (No Response)

Reviewer #2: (No Response)

Reviewer #3: (No Response)

**Summary and General Comments**

Reviewer #1: (No Response)

Reviewer #2: (No Response)

Reviewer #3: My comments have been adequately addressed in the revised manuscript

PLOS authors have the option to publish the peer review history of their article (what does this mean?). If published, this will include your full peer review and any attached files.

Reviewer #1: **Yes: **Mohammad Ali

Reviewer #2: No

Reviewer #3: No

---

## [Editor Report · Acceptance letter]

27 Apr 2023

Dear Dr Charnley,

We are delighted to inform you that your manuscript, "Cholera past and future in Nigeria: are the Global Task Force on Cholera Control’s 2030 targets achievable?," has been formally accepted for publication in PLOS Neglected Tropical Diseases.

Best regards,

Shaden Kamhawi

co-Editor-in-Chief

Paul Brindley

co-Editor-in-Chief
